# Effect of *Micromelalopha sieversi* (Staudinger) Oviposition Behavior on the Transcriptome of Two *Populus* Section *Aigeiros* Clones

**Li Guo** [1,2], **Sufang Zhang** [1], **Fu Liu** [1], **Xiangbo Kong** [1] **and Zhen Zhang** [1,*]

1    Research Institute of Forest Ecology, Environment and Protection, Chinese Academy of Forestry,
     Beijing 100091, China; fangziyibaobei@126.com (L.G.); zhangsf@caf.ac.cn (S.Z.); liufu2006@163.com (F.L.);
     xbkong@sina.com (X.K.)
2    School of Biological Science and Engineering, Xingtai University, Xingtai 054001, China
*    Correspondence: zhangzhen@caf.ac.cn; Tel.: +86-136-7102-2209

**Abstract:** *Research Highlights:* The molecular mechanisms underlying woody plant resistance upon oviposition by herbivores remain unclear, as studies have focused on herbaceous plants. The effect of oviposition on gene expression in neighboring plants has also not been reported. Elucidating these molecular responses can help cultivate insect-resistant trees. *Background and Objectives*: Oviposition by herbivorous insects acts as an early warning signal, inducing plant resistance responses. Here, we employed poplar as a model woody plant to elucidate gene expression and the molecular mechanisms underlying plant resistance after oviposition by *Micromelalopha sieversi* (Staudinger) (Lepidoptera: Notodontidae). *Materials and Methods:* The differences in gene expression of two *Populus* section *Aigeiros* clones ('108' (*Populus* × *euramericana* 'Guariento') and '111' (*Populus* × *euramericana* 'Bellotto')) were analyzed via high-throughput sequencing of oviposited, neighboring, and control plants. *Results:* We obtained 304,526,107 reads, with an average length of 300 bp and a total size of 40.77 Gb. Differentially expressed genes (DEGs) in gene ontology terms of biological process, cellular component, and molecular function were mainly enriched in the "cell part", "catalytic", and "metabolic process" functions. Moreover, DEGs were mainly enriched in the following pathways: plant-pathogen interaction, linoleic acid metabolism, and cyanoamino acid metabolism (108-O vs. 108-C); metabolic pathways, photosynthesis, photosynthesis-antenna proteins, nitrogen metabolism, and linoleic acid metabolism (111-O vs. 111-C); metabolic pathways and biosynthesis of secondary metabolites (111-N vs. 111-C); no pathways were significantly enriched in 108-N vs. 108-C. Up-regulated defense genes were associated with pathogenesis-related protein function, innate immune regulation, and biological stress response, with differences in specific genes. All genes related to photosynthetic activity were significantly down-regulated in oviposited and neighboring leaves of the two clones. *Conclusions:* Oviposited and neighboring '108' and '111' plants exhibited varying degrees of resistance upon oviposition, involving the up-regulation of various defense genes, decreased photosynthesis and nutrient accumulation, and increased secondary metabolic synthesis.

**Keywords:** woody plant; clones; *Micromelalopha sieversi* (Staudinger); oviposition; neighboring plants; differentially expressed genes; induced resistance

## 1. Introduction

The induced resistance of plants constitutes physiological, biochemical, and morphological changes following exposure to phytophagous insects [1]. The direct defenses include the activation of defense gene expression; metabolic processes that reconfigure to produce toxic compounds,

antinutrient enzymes, and antidigestive enzymes [2]. As such, the utilization of plant induced resistance to reduce the losses resulting from pests or diseases would be an ideal strategy for ecological forestry and environmental protection [3]. Oviposition by phytophagous insects can be used as an early damage signal to induce plant resistance directly or indirectly [4,5]. Oviposition can cause changes in the photosynthesis and the release of volatiles and chemical substances on the leaf surface of damaged plants [6]; for example, *Pinus sylvestris* (L.) reduces photosynthetic activity after oviposition by *Diprion pini* (L.) (Hymenoptera: Diprionidae) [7], while egg deposits alter the proportion of waxy compounds in the leaves of *Arabidopsis* [8]. Oviposition can also induce changes in the volatile release of damaged plants, mainly in terms of quantity [9,10]. Notably, egg-free leaves neighboring egg-laden ones also show systemically induced resistance [11–13].

The expression of plant genes such as those involved in the biosynthesis of antimycin [14], glucosinolate [15], protease inhibitor (PIN) [16], or terpene [17–19] may be affected with the oviposition of insect eggs. In tomato, a strong induction of the defense gene *PIN2*, encoding an anti-insect proteinase inhibitor [20], was observed under and in the vicinity of *Helicoverpa zea* (Boddie) (Lepidoptera: Noctuidae) eggs [16]. Moreover, the expression of hundreds of genes was altered over a period of three days after oviposition by *Pieris brassicae* (L.) (Lepidoptera: Pieridae) on *Arabidopsis thaliana*, including those encoding defense proteins, regulators of cell death and innate immunity, genes responding to biotic and abiotic stresses, and genes associated with the production of secondary metabolites involved in plant defense [21]. Another study also demonstrated that oviposition induced the expression of many defense-related genes, including genes encoding pathogenesis-related (PR) proteins (chitinases and glucanases) and genes involved in abiotic stress and phytohormone signaling [22].

Salicylic acid (SA), an important signal that mediates defense against various pathogens, is both essential and sufficient for inducing systemic acquired resistance (SAR) [23]. Jasmonic acid (JA) and its derivatives are involved in an intracellular signal cascade that begins with the interaction of an elicitor molecule with the plant cell surface and results in the accumulation of secondary compounds [24]. SA and JA induce the production of antimicrobial compounds such as phytoalexins and pathogenesis-related (PR) proteins [25]. Eggs may be perceived as microbial pathogens, at least at the molecular level [26]. The reaction of oviposited leaves confirms this, as SA was found accumulated at high levels underneath the egg mass and many SA-responsive genes are induced by oviposition [21,27]. In addition, the importance of the SA pathway in response to egg-derived elicitors in *A. thaliana* has been demonstrated [28]. Meanwhile, oviposition by *Xanthogaleruca luteola* (L.) (Coleoptera: Chrysomelidae) in elm and pea pods induces JA biosynthesis genes [4,22]. The JA pathway appears to be prominent in cases where oviposition is accompanied by wounding of the leaf, whereas the SA pathway was shown to be involved when eggs are only deposited onto the surface without any apparent damage [26].

Previous studies have researched how herbivorous insects lay eggs as an early warning signal, inducing the plant resistance response; however, they mostly focus on herbaceous plants. There are only a few studies on the induced resistance of forest trees caused by phytophagous insect oviposition. Moreover, it has not been reported if oviposition produces induced resistance and genetic changes to neighboring poplar plants. *Populus* is widely accepted as a model tree for forest genome research, and its full genome sequence and annotation information is available on the Joint Genome Institute; a genome of *Populus trichocarpa* (Torr. and Gray) was published in 2006 [29]. There have also been a few studies on poplar gene expression; for example, the mechanism of heterosis formation of *Populus deltoides* (Marsh) was revealed via high-throughput sequencing and analysis of differentially expressed genes (DEGs) [30]. *Micromelalopha sieversi* (Staudinger) (Lepidoptera: Notodontidae) is a serious popular pest that is difficult to control. Previous studies have found that oviposition by *M. sieversi* on '108' (*Populus × euramericana* 'Guariento') and '111' (*Populus × euramericana* 'Bellotto') induced resistance in oviposited plants and neighboring plants, and that there were differences between the two *Populus* section *Aigeiros* clones [31]. The purpose of this experiment was to investigate the gene expression of two healthy clones ('108' and '111'), as well as the alterations in gene expression

caused by oviposition; plants that received the eggs, as well as the neighboring plants were analyzed. In this study, we analyzed the gene expression of the two *Populus* section *Aigeiros* clones to reveal the difference of insect-resistance between the two clones and the induced resistance mechanisms in oviposited and neighboring plants after oviposition by *M. sieversi*. Our findings provide insights into these molecular responses for elucidating woody plant resistance induced by herbivore oviposition and the cultivation of new insect-resistant poplars.

## 2. Materials and Methods

### 2.1. Plants and Insects

In 2015, branches of '108' (*Populus × euramericana* 'Guariento') and '111' (*P. × euramericana* 'Bellotto') were cut and collected from the nursery of the Chinese Academy of Forestry. Each branch was cut into small segments (about 12–13 cm) with 3–4 buds and cultivated outside for 3–10 months in separate plastic pots filled with nutrient-containing soil. Plants, 4–5 months old, with more than a dozen expanded leaves, were transferred to a laboratory [26 ± 2 °C, 50% ± 5% relative humidity, 16:8 h light: dark] and used for oviposition experiments. *Micromelalopha sieversi* pupae were collected from Feilou in the Fengxian county of Xuzhou (China), after which the female and male pupae were placed separately and hatched naturally.

### 2.2. Plant Treatments

Experiments were carried out in two laboratories under identical natural conditions. The plants were divided into three treatment groups of oviposited plants, plants neighboring the oviposited plants, and control plants. Two insect cages (0.75 m × 0.75 m × 1.5 m) for the four oviposited plants and the four neighboring plants were set up and all plants were placed approximately 50 cm apart; the other insect cage for the four control plants was set up in the second laboratory. Newly hatched male and female moths (20 pairs) were free to mate and lay eggs in the cage of oviposited plants. Towards the end of the egg incubation period (72 h), leaves with eggs were collected (to remove the egg mass), and the corresponding parts of the leaves from the neighboring and control plants were collected. All leaves were stored in liquid nitrogen for RNA extraction. The experiment was repeated three times with the '108' and '111' clones.

### 2.3. RNA Quality Detection and cDNA Library Preparation

Total RNA was extracted from the leaves using TRIzol Reagent (Invitrogen, Carlsbad, CA, USA), after which RNA integrity was determined with the Agilent 2100 RNA Nano 6000 Assay Kit (Agilent Technologies, Santa Clara, CA, USA). Purified double-stranded cDNA was enriched by PCR to obtain the final cDNA library. After the library was built, HiSeq PE Cluster Kit v4-cbot HS (Illumina, San Diego, CA, USA) reagent was used to generate clusters on cBot. Then, the dual-end sequencing program was run on the HiSeq4000 sequencing platform and 150 bp paired-end sequencing reads were obtained.

### 2.4. Sequence Alignment and Analysis of Differential Gene Expression

Filtered data were compared with the reference genome database of *Populus trichocarpa* [29]. FPKM (reads per kb per million reads) [32] were obtained for each sample to calculate gene expression. DESeq was used for differential expression analysis. After correction, the *P* value and log2 (fold change) thresholds were set as 0.05 and 1, respectively. By comparing the treatment group and the reference group, genes with a log2 ratio of ≥1 and q ≤ 0.05 were selected as DEGs, and the number of up- and down-regulated genes was obtained. Comparisons were made using the genomic annotation database of *Populus trichocarpa* [29]. DEGs across the three '108' and '111' treatment groups were stratified and clustered using R software (version v3.1.1) [33]. DEGs were annotated using Uniprot and the NCBI NT, NR, and COG databases, as well as gene ontology (GO) and the Kyoto Encyclopedia of

Genes and Genomes (KEGG) databases. The main biological functions of DEGs were determined by GO significant enrichment analysis. Metabolic pathway annotation of DEGs was performed using KEGG, and the enrichment degree of DEGs was calculated using the Fisher test with classic Fisher ≤0.01 as the screening threshold.

## 3. Results

### 3.1. Transcriptome Sequencing and Reference Genome Comparisons of Each Treatment

A total of 304,526,107 sequences (average length: 300 bp; size: ~40.77 Gb) were obtained from the cDNA libraries constructed from the six treatments (oviposited leaves of '108'(108-O), neighboring leaves of '108'(108-N), control leaves of '108'(108-C), oviposited leaves of '111'(111-O), neighboring leaves of '111'(111-N), control leaves of '111'(111-C)) (Table 1); 50,854,144 sequences were produced per treatment on average, and the total number of high-quality sequences per treatment accounted for 89.20% of the total number of original sequences on average. Moreover, 63.60% of sequences per treatment could be matched to the reference genome, while 36.36% could not be matched on average; meanwhile, 3.07% were matched to multiple locations on the genome. The number and proportion of unique alignment sequences to the three functional components of the gene (exon, intron, and intergenic region) were compared to obtain the distribution of each sequence on the genes of different treatment groups (Table 2). The average ratios of the unique alignment sequences in exons, introns, and intergenic regions were 93.63%, 4.06%, and 2.30%, respectively. The unique alignment sequence was used for subsequent gene expression analysis.

**Table 1.** Sequencing data and genome mapping analysis.

|  | 108-O | 108-N | 108-C | 111-O | 111-N | 111-C |
|---|---|---|---|---|---|---|
| Raw reads number | 50,255,389 | 52,672,759 | 52,355,816 | 51,052,837 | 49,999,297 | 48,190,009 |
| Clean reads number | 45,098,385 (89.74%) | 46,656,841 (88.58%) | 46,289,664 (88.43%) | 45,536,401 (89.20%) | 44,735,416 (89.50%) | 43,503,490 (90.28%) |
| Mapped reads | 28,247,431 (62.67%) | 29,579,017 (63.33%) | 29,966,665 (64.67%) | 28,427,433 (62.33%) | 28,158,929 (63.00%) | 28,140,356 (64.67%) |
| Unmapped reads | 16,850,954 (37.36%) | 17,077,825 (36.60%) | 16,322,999 (35.26%) | 17,108,969 (37.57%) | 16,576,487 (37.05%) | 15,363,134 (35.31%) |
| Multi map reads | 1,061,769 (2.67%) | 1,375,346 (3.00%) | 1,967,076 (4.33%) | 1,355,164 (3.00%) | 893,508 (2.00%) | 1,130,265 (3.00%) |

**Table 2.** Comparison read distribution statistics.

|  | 108-O | 108-N | 108-C | 111-O | 111-N | 111-C |
|---|---|---|---|---|---|---|
| Exon | 13,453,196 (93.68%) | 13,482,984 (93.03%) | 13,358,075 (94.08%) | 13,327,758 (93.84%) | 12,615,311 (92.46%) | 13,056,880 (94.72%) |
| Intron | 581,278 (4.05%) | 648,371 (4.49%) | 514,929 (3.63%) | 552,302 (3.91%) | 700,806 (5.07%) | 443,171 (3.23%) |
| Intergenic | 326,553 (2.27%) | 360,234 (2.49%) | 514,929 (2.29%) | 320,559 (2.26%) | 337,833 (2.47%) | 282,035 (2.05%) |

### 3.2. Gene Expression Analysis of the Different Treatment Groups

The gene expression density map of all samples, which followed a bimodal distribution, was obtained (Figure 1). The three treatment groups of '108' exhibited 29,687 co-expression genes, with 417, 645, and 751 specific genes expressed in control, neighboring, and oviposited plants, respectively (Figure 2A). The three treatment groups of '111' had 29,187 co-expression genes, with 976, 372, and 567 specific genes expressed in control, neighboring, and oviposited plants, respectively (Figure 2B). Meanwhile, the control, neighboring, and oviposited '108' and '111' plants shared 30,130, 29,982, and 30,112 co-expression genes, respectively, whereas the specific genes expressed in '108' and

'111' amounted to 799 and 1347 in the control, 1798 and 683 in the neighboring, and 1541 and 760 in the oviposited plants, respectively (Figure 2C).

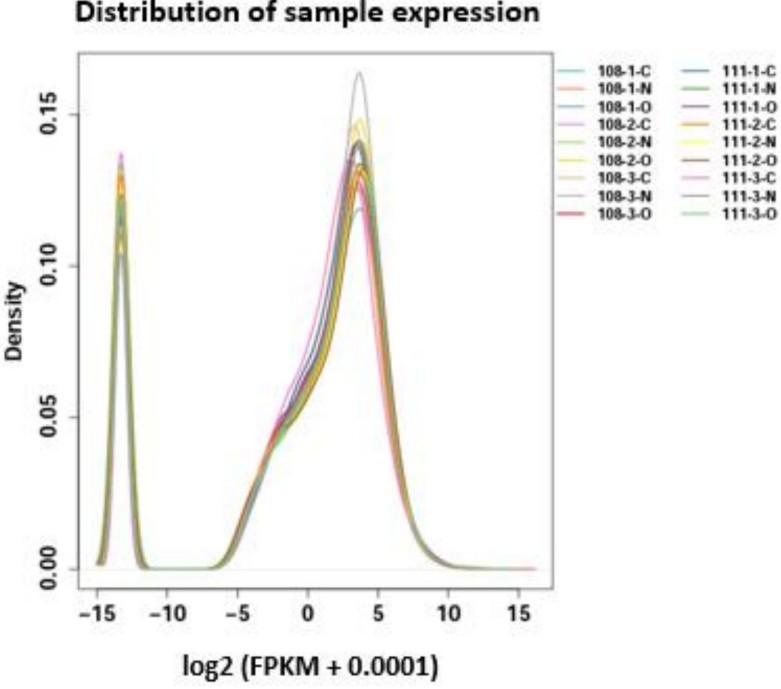

**Figure 1.** Gene expression density map of '108' and '111' treatment groups. The different samples are represented by different colors.

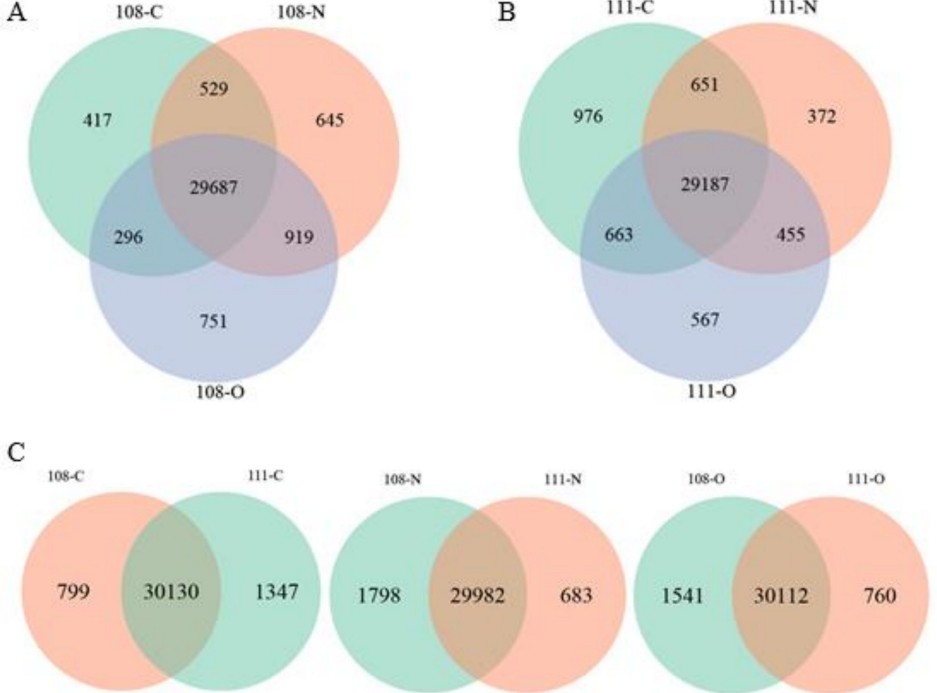

**Figure 2.** Gene expression analysis of the '108' and '111' treatment groups. Venn diagrams representing the gene expression of (**A**) control, neighboring, and oviposited leaves of '108', (**B**) control, neighboring, and oviposited leaves of '111', and (**C**) control, neighboring, and oviposited leaves of '108' and '111.' C, control; N, neighboring; O, oviposited.

### 3.3. Analysis of DEGs

DEG analysis of '108' revealed 571 DEGs (408 up-regulated and 163 down-regulated) between oviposited and control leaves (108-O vs. 108-C), 48 (16 up-regulated and 32 down-regulated) between neighboring and control leaves (108-N vs. 108-C), and 283 (244 up-regulated and 39 down-regulated) between oviposited and neighboring leaves (108-N vs. 108-C) (Figure 3A). For clone '111', 1234 genes were differently expressed (433 up-regulated and 801 down-regulated) between oviposited and control leaves (111-O vs. 111-C), 1784 (173 up-regulated and 1611 down-regulated) between neighboring and control leaves (111-N vs. 111-C), and 646 genes (606 up-regulated and 40 down-regulated) between oviposited and neighboring leaves (111-N vs. 111-C) (Figure 3B). Meanwhile, a comparison of the '111' and '108' clones revealed 43 DEGs (6 up-regulated and 37 down-regulated) in the oviposited leaves (111-O vs. 108-O), 645 (14 up-regulated and 631 down-regulated) in the neighboring leaves (111-N vs. 108-N), and 24 (10 up-regulated and 14 down-regulated) in control leaves (111-C vs. 108-C) (Figure 3C).

Clustering analysis of '108' showed that DEGs in the control and neighboring leaves were clustered together, while those of the neighboring and oviposited leaves were clustered together; this indicates that the defense response of '108' after oviposition was continuously changing, with no significant difference between the three treatments (Figure 4A). For '111', the control group DEGs were clustered together; those of the neighboring and oviposited leaves were clustered together, indicating that neighboring plants rapidly detected the stress of oviposited plants and produced similar defensive responses (Figure 4B).

### 3.4. GO Functional Enrichment Analysis of DEGs

GO analysis revealed that 2830 genes in the oviposited leaves of '108' compared with those in the control could be functionally annotated and that they were mainly involved in 42 biological functions. GO terms of biological process, cellular component, and molecular function showed 19, 11, and 12 enriched biological functions, respectively, including "cell part" with the highest enrichment (218 up-regulated and 90 down-regulated), followed by "catalytic" (183 up-regulated and 77 down-regulated) and "metabolic process" (167 up-regulated and 80 down-regulated) (Figure 5A). When the '108' neighboring leaves were compared with the control, 276 genes could be functionally annotated and were found mainly involved in 30 biological functions. The biological process, cellular component, and molecular function GO terms showed 11, 9, and 10 enriched biological functions, respectively. The "cell part" function was the most enriched (11 up-regulated and 19 down-regulated), followed by "single-organism process" (7 up-regulated and 17 down-regulated) and "catalytic" (3 up-regulated and 20 down-regulated) (Figure 5B).

For clone '111', 6310 genes in the oviposited leaves compared with those in the control could be functionally annotated and were found mainly involved in 44 biological functions. The GO terms biological process, cellular component, and molecular function showed 20, 12, and 12 enriched biological functions, respectively. The "cell part" function was again the most enriched (227 up-regulated and 513 down-regulated), followed by "cellular process" (172 up-regulated and 392 down-regulated) and "binding" (213 up-regulated and 339 down-regulated) (Figure 5C). Finally, 9338 genes in the '111' neighboring leaves compared with those in the control could be functionally annotated and were found mainly involved in 46 biological functions. Biological process, cellular component, and molecular function GO terms showed 20, 12, and 14 enriched biological functions, respectively. The most enriched function was the "cell part" (100 up-regulated and 1021 down-regulated), followed by "metabolic process" (76 up-regulated and 775 down-regulated) and "catalytic" (86 up-regulated and 707 down-regulated) (Figure 5D).

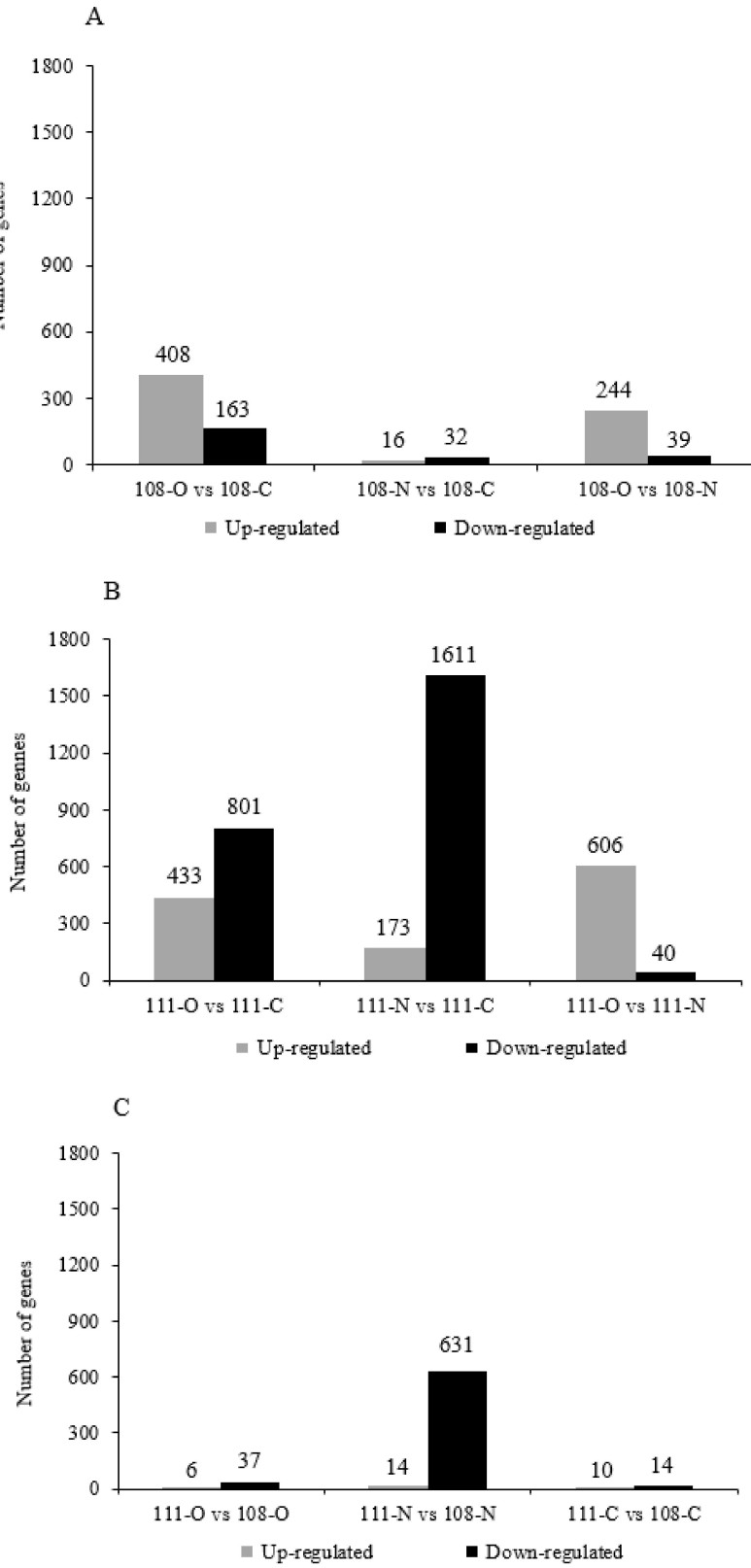

**Figure 3.** Differential gene expression analysis of the '108' and '111' treatment groups. Differentially expressed genes (DEGs) among (**A**) oviposited, neighboring, and control leaves of '108', (**B**) oviposited, neighboring, and control leaves of '111', and (**C**) oviposited, neighboring, and control leaves of '111' and '108'. C, control; N, neighboring; O, oviposited.

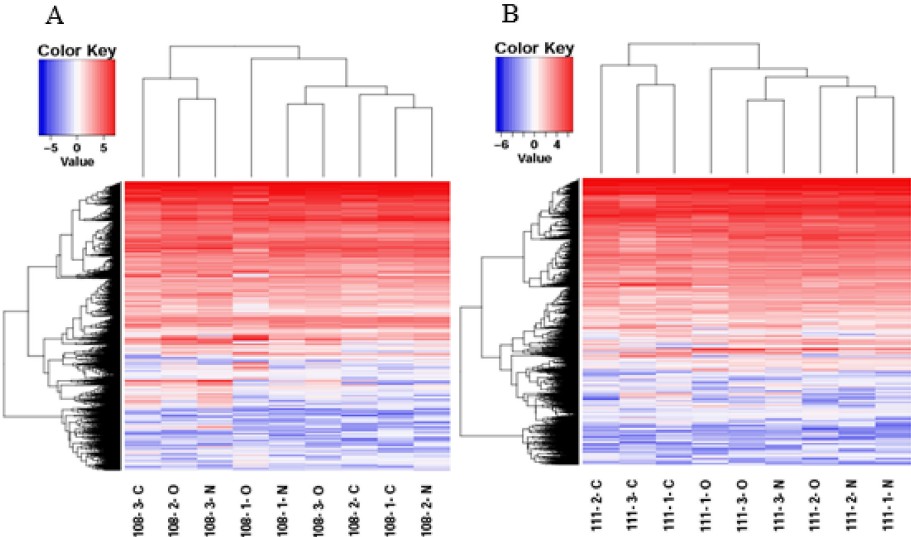

**Figure 4.** DEG dendrograms of the '108' and '111' treatment groups. Dendrograms showing the clustering of DEGS in the (**A**) '108' and (**B**) '111' treatment groups. Pearson correlation co-efficient between pairwise, and pairwise was calculated according to the gene expression of each sample to indicate the similarity between the two treatments. Using system clustering (i.e., hierarchical cluster) to classify and group samples with high similarity, the clustering results of '108' and '111' were obtained. C, control; N, neighboring; O, oviposited.

*3.5. Pathway Enrichment Analysis of DEGs*

Compared to the DEGs in the control leaves, a large number of DEGs in the oviposited '108' leaves were mainly enriched within three functional pathways, with the plant–pathogen interaction representing the highest enrichment, followed by the linoleic acid metabolism and cyanoamino acid metabolism pathways (Figure 6A). These results indicate that the oviposited plants of '108' sensed an invasion and up-regulated their defense through interactions between plant and pathogenic bacteria after oviposition by *M. sieversi*. DEGs in the neighboring '108' plants were not significantly enriched in metabolic pathways compared with those of the control; however, the genes involved in porphyrin and chlorophyll metabolism as well as the starch and sucrose metabolism pathways were down-regulated, while those involved in cutin, suberine, and wax biosynthesis and the plant-pathogen interaction pathway were up-regulated. This indicates that the physiological activities of the neighboring '108' plants were affected and that part of the defense response was activated.

Meanwhile, a large number of DEGs in the oviposited '111' leaves were mainly enriched in metabolic pathways, photosynthesis, photosynthesis-antenna, nitrogen metabolism, and linoleic acid metabolism pathways compared with those in the control (Figure 6B). Thus, photosynthesis of '111' was affected by oviposition. The neighboring '111' plants when compared with control showed that the majority of DEGs were enriched in 13 functional pathways, with metabolic pathways and biosynthesis of secondary metabolites showing the highest enrichment (Figure 6C). These results indicate that physiological metabolic processes of neighboring plants were affected by oviposition and that generation of secondary metabolites is also a sign of plant resistance against insects.

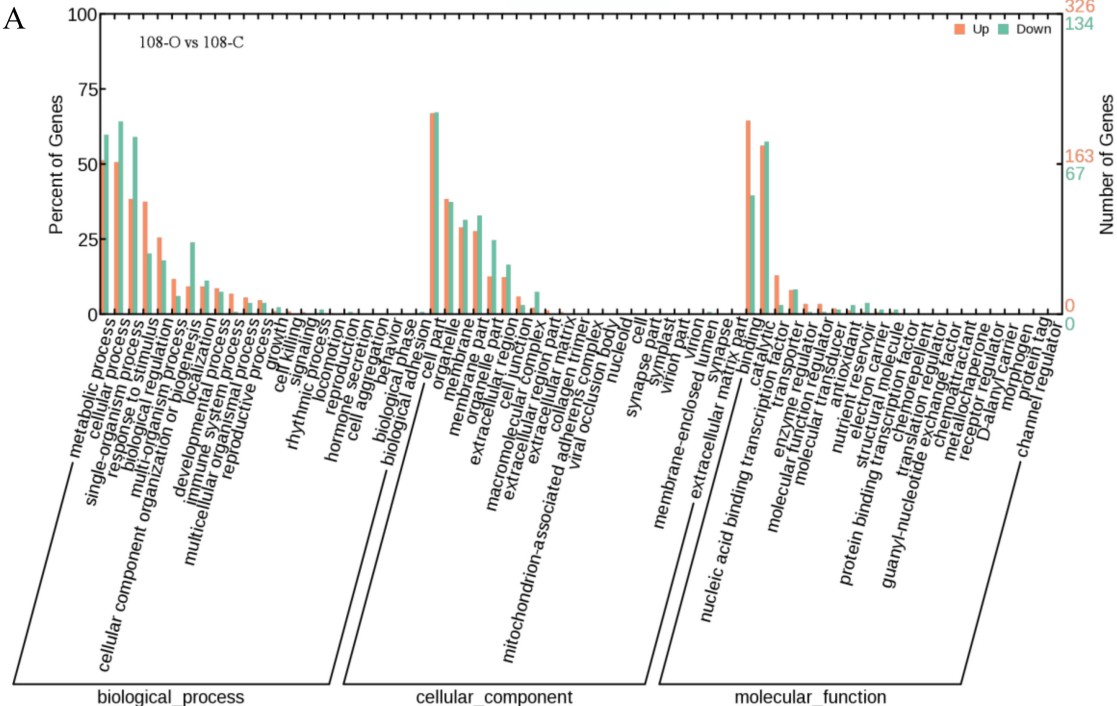

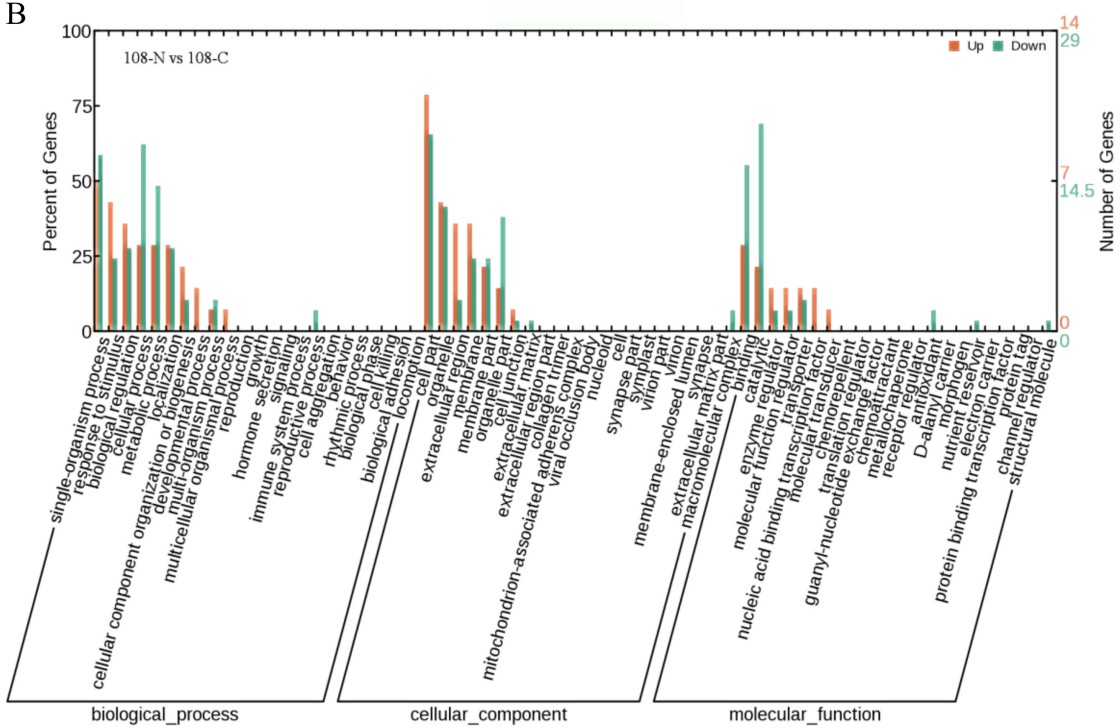

**Figure 5.** *Cont.*

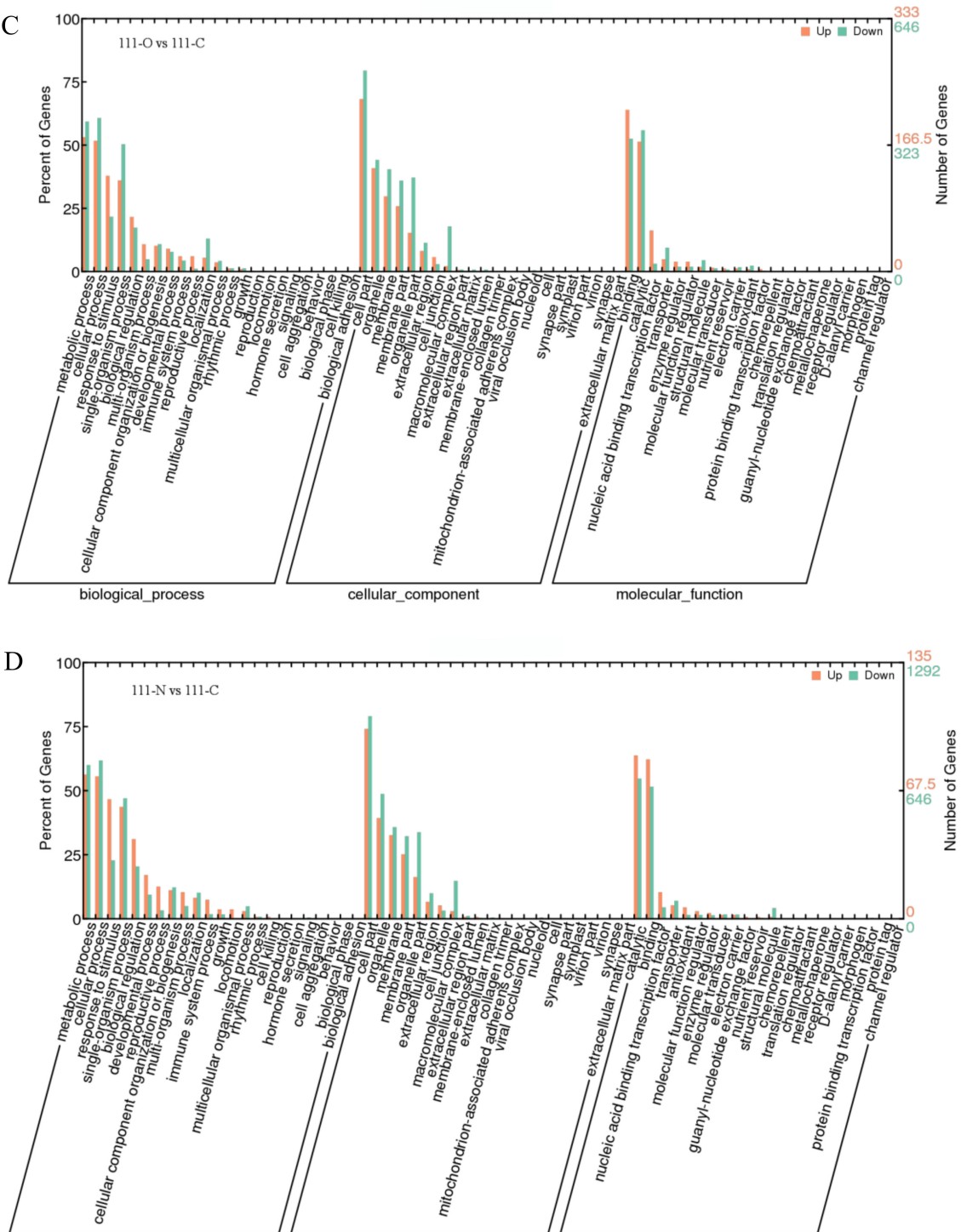

**Figure 5.** Gene ontology (GO) analysis of the DEGs among the '108' and '111' treatment groups. GO analysis of the DEGs between (**A**) oviposited and control leaves of '108,' (**B**) neighboring and control leaves of '108,' (**C**) oviposited and control leaves of '111,' and (**D**) neighboring and control leaves of '111'. C, control; N, neighboring; O, oviposited; up, up-regulated genes; down, down-regulated genes.

A

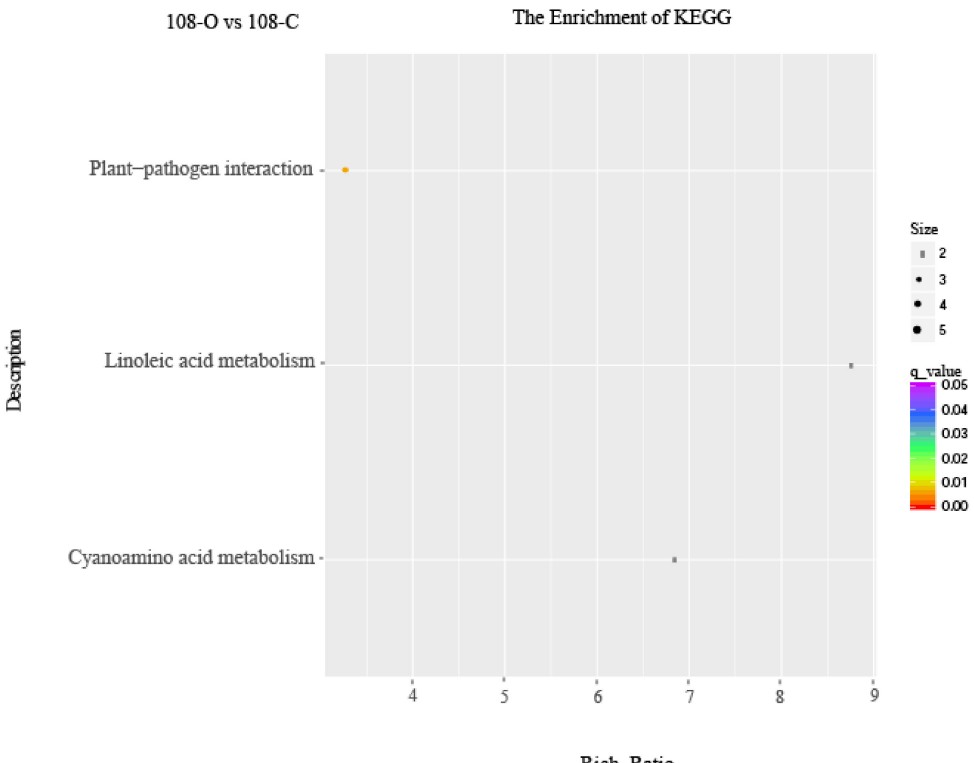

B

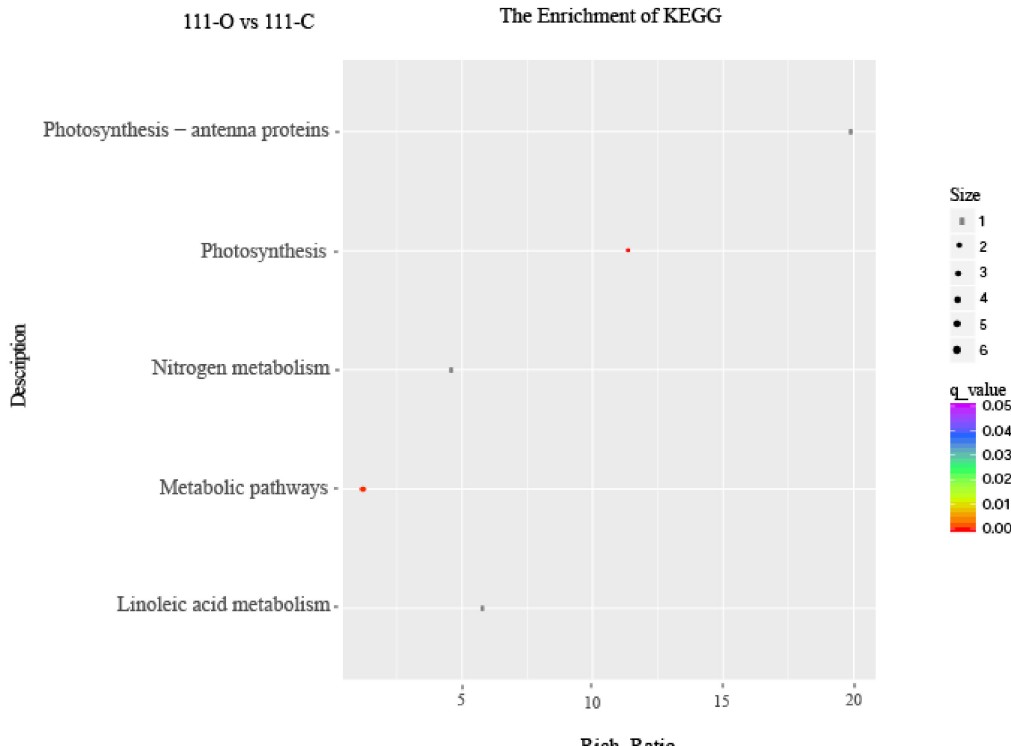

**Figure 6.** *Cont.*

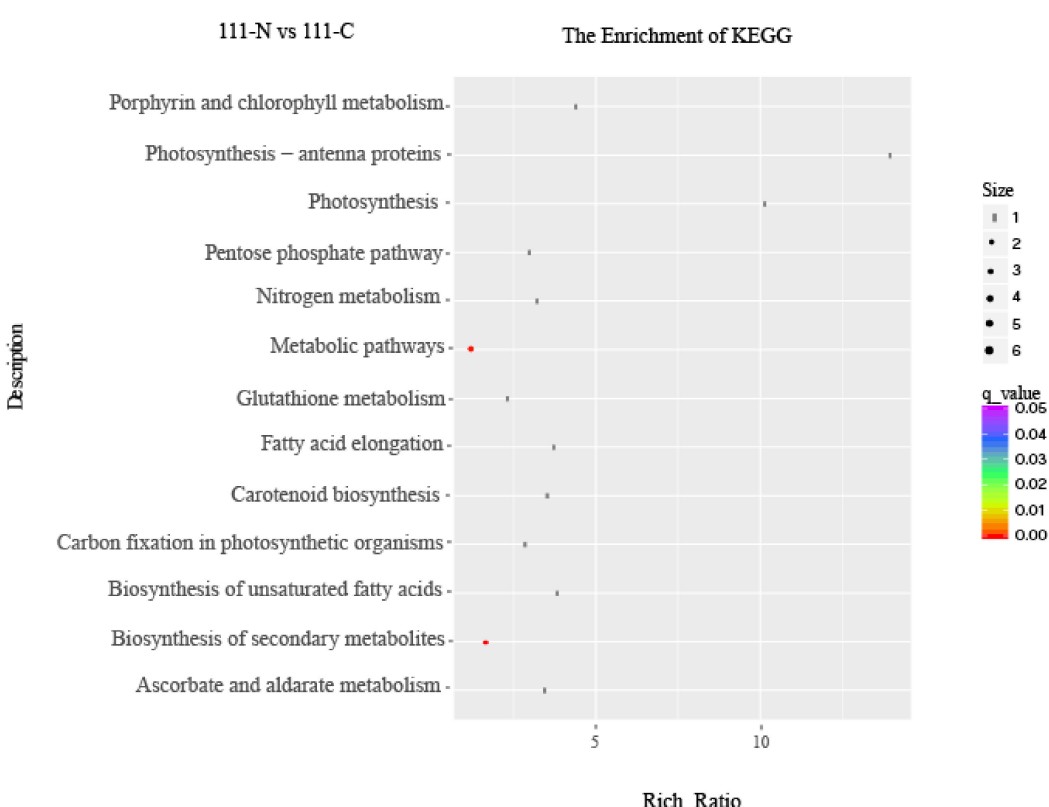

**Figure 6.** Kyoto Encyclopedia of Genes and Genomes (KEGG) pathway analysis of DEGs among the '108' and '111' treatment groups. DEG pathway enrichment between (**A**) oviposited and control leaves of '108', (**B**) oviposited and control leaves of '111,' and (**C**) neighboring and control leaves of '111'. Each point indicates the degree of enrichment of the KEGG entry. The closer the color is to red, the higher the degree of enrichment. The size of each point indicates the number of genes that are enriched in the KEGG entry, and the larger the point, the more genes that are enriched.

*3.6. Differential Expression of Important Defense Genes*

Important defense-related genes in the oviposited leaves of '108' that were significantly up-regulated included genes related to the following: PR proteins such as trypsin and protease inhibitor (TPIN) (POPTR_0001s31740g); innate immune regulation such as NPR1/NIM1-interacting protein 2 (NIMIN-2) (POPTR_0002s19140g) and the mildew resistance protein RPW8 homolog (HR4) (POPTR_0007s115001g), which is associated with biological stress response; proteolysis, such as metalloproteinase (POPTR_0012s10640g) and L-ascorbate oxidase (POPTR_0005s08120g); and cell wall metabolism, such as xyloglucan endotransglycosylase (XTR8) (POPTR_0001s01870g). For the neighboring leaves, significantly up-regulated genes included biostress-related genes such as pathogen-responsive α-dioxygenase (POPTR_0012s04690g) ($P = 0.0072$); these levels were significantly higher than those of oviposited leaves ($P = 0.0066$). Significantly down-regulated genes in oviposited and neighboring leaves were associated with photosynthetic activity, such as PSII 5-kD protein (POPTR_0001s26410g), PSII core complex PsbY (PSBY) (POPTR_0008s18230g), PSII reaction center PsbP (POPTR_0010s21710g), PSII assembly factor HCF136 (POPTR_0007s07780g), oxygen evolving enhancer 3 (PsbQ) (POPTR_0010s17400g), ferredoxin (POPTR_0007s09630g), and ferredoxin-thioredoxin reductase (POPTR_0001s34690g). Moreover, the PSII reaction center PsbW (POPTR_0002s25810g) and divinyl protochlorophyllide a 8-vinyl reductase (PCB2) (POPTR_0008s20940g) were significantly down-regulated in oviposited leaves ($P = 0.0332$), whereas the PSI reaction center subunit VI

(POPTR_0003s05110g) and chlorophyll a/b-binding protein (LHCB3) (POPTR_0002s22220g) were significantly down-regulated in the neighboring leaves (*P* = 0.0079) (Table 3).

For the oviposited leaves of '111', important defense-related genes that were significantly up-regulated included genes related to the following: PR protein such as the hevein-like protein (PR4/HEL) (POPTR_0013s03890g); innate immune regulation, such as the phytoalexin-deficient 4 protein/lipase (PAD4) (POPTR_0005s06960g); cell death, such as BON1-associated protein 1 (BAP1) (POPTR_0014s07470g); and cell wall metabolism, such as XTR8. For the neighboring leaves, XTR8 (*P* = 0.0017) and cytochrome P450 (POPTR_0002s02700g) (*P* = 0.0175) related to the tryptophan pathway were significantly up-regulated, while BAP1 was significantly down-regulated (*P* = 0.0154). Common genes significantly down-regulated in oviposited and neighboring leaves included genes associated with photosynthetic activity, such as the PSI reaction center subunit VI, PSII 5-kD protein, PSBY, PsbP, PsbW, PSII assembly factor HCF136 (POPTR_0007s07780g), PCB2, LHCB3, PsbQ, cytochrome c biogenesis protein (POPTR_0009s11490g), ferredoxin, and ferredoxin-thioredoxin reductase; genes associated with the Calvin cycle, such as fructose-1,6-bisphosphatase (POPTR_0005s21360g); and genes related to starch synthesis, such as starch synthase (POPTR_0011s15540g) (Table 4).

**Table 3.** Differential gene expression of important defense genes among the '108' treatment groups.

| Gene Description | Gene Symbol | Gene ID | 108-O vs. 108-C | | 108-N vs. 108-C | | 108-O vs. 108-N | |
|---|---|---|---|---|---|---|---|---|
| | | | Ratio | *P* Value | Ratio | *P* Value | Ratio | *P* Value |
| Trypsin and protease inhibitor | TPIN | POPTR_0001s31740g | 52.77 | 0.0003 | 4.32 | 0.6394 | 12.24 | 0.0081 |
| NPR1/NIM1-interacting protein 2 | NIMIN-2 | POPTR_0002s19140g | 577.01 | 0.0000 | 33.21 | 0.0592 | 17.56 | 0.0457 |
| Mildew resistance protein, RPW8 homolog | HR4 | POPTR_0007s115001g | 3.99 | 0.0377 | 1.10 | 0.8886 | 3.65 | 0.0537 |
| Pathogen-responsive α-dioxygenase | | POPTR_0012s04690g | 8.10 | 1.0000 | 95.76 | 0.0072 | 0.08 | 0.0066 |
| Metalloproteinase | | POPTR_0012s10640g | 9.19 | 0.0001 | 1.41 | 0.6673 | 6.58 | 0.0135 |
| L-Ascorbate oxidase | | POPTR_0005s08120g | 31.85 | 0.0000 | 0.57 | 0.7684 | 55.79 | 0.0000 |
| Xyloglucan endotransglycosylase | XTR8 | POPTR_0001s01870g | 17.67 | 0.0010 | 1.69 | 0.6164 | 10.44 | 0.0322 |
| PSI reaction center subunit VI | | POPTR_0003s05110g | 0.25 | 0.0604 | 0.42 | 0.0083 | 0.59 | 0.2488 |
| PSII 5-kD protein | | POPTR_0001s26410g | 0.17 | 0.0209 | 0.42 | 0.0072 | 0.41 | 0.0847 |
| PSII core complex PsbY | PSBY | POPTR_0008s18230g | 0.30 | 0.0241 | 0.53 | 0.0412 | 0.57 | 0.1884 |
| PSII reaction center PsbP | | POPTR_0010s21710g | 0.27 | 0.0272 | 0.35 | 0.0021 | 0.76 | 0.6710 |
| PSII reaction center PsbW | | POPTR_0002s25810g | 0.17 | 0.0029 | 0.48 | 0.1582 | 0.36 | 0.0892 |
| PSII assembly factor HCF136 | HCF136 | POPTR_0007s07780g | 0.23 | 0.0038 | 0.33 | 0.0015 | 0.71 | 0.4393 |
| Divinyl protochlorophyllide a 8-vinyl reductase | PCB2 | POPTR_0008s20940g | 0.28 | 0.0332 | 0.54 | 0.0872 | 0.51 | 0.2402 |
| Chlorophyll a/b-binding protein | LHCB3 | POPTR_0002s22220g | 0.24 | 0.0780 | 0.42 | 0.0079 | 0.58 | 0.2528 |
| Oxygen evolving enhancer 3 (PsbQ) | | POPTR_0010s17400g | 0.20 | 0.0004 | 0.42 | 0.0213 | 0.48 | 0.0944 |
| Ferredoxin | | POPTR_0007s09630g | 0.13 | 0.0197 | 0.31 | 0.0098 | 0.44 | 0.3389 |
| Ferredoxin-thioredoxin reductase | | POPTR_0001s34690g | 0.33 | 0.0442 | 0.45 | 0.0155 | 0.74 | 0.5784 |

Ratio ≥2 or ≤0.5 and *P* value < 0.05 indicate a significant difference.

**Table 4.** Differential gene expression of important defense genes among the '111' treatment groups.

| Gene Description | Gene Symbol | Gene ID | 111-O vs. 111-C | | 111-N vs. 111-C | | 111-O vs. 111-N | |
|---|---|---|---|---|---|---|---|---|
| | | | Ratio | *P* Value | Ratio | *P* Value | Ratio | *P* Value |
| Hevein-like protein | PR4/HEL | POPTR_0013s03890g | 2.70 | 0.0274 | 1.20 | 0.5848 | 2.27 | 0.3236 |
| Phytoalexin-deficient 4 protein/lipase | PAD4 | POPTR_0005s06960g | 5.35 | 0.0026 | 1.76 | 0.4008 | 3.04 | 0.0031 |
| BON1-associated protein 1 | BAP1 | POPTR_0014s07470g | 10.39 | 0.0285 | 0.18 | 0.0154 | 58.56 | 0.0000 |
| Xyloglucan endotransglycosylase | XTR8 | POPTR_0001s01870g | 13.70 | 0.0000 | 5.40 | 0.0017 | 2.53 | 0.0016 |
| Cytochrome P450 | CYP83B1 | POPTR_0002s02700g | 1.48 | 0.4434 | 2.70 | 0.0175 | 0.54 | 0.1408 |
| PSI reaction center subunit VI | | POPTR_0003s05110g | 0.19 | 0.0000 | 0.24 | 0.0000 | 0.79 | 0.3069 |
| PSII 5-kD protein | | POPTR_0001s26410g | 0.13 | 0.0000 | 0.21 | 0.0000 | 0.65 | 0.1129 |
| PSII core complex PsbY | PSBY | POPTR_0008s18230g | 0.29 | 0.0017 | 0.29 | 0.0000 | 1.00 | 0.9721 |
| PSII reaction center PsbP | | POPTR_0010s21710g | 0.22 | 0.0000 | 0.35 | 0.0079 | 0.63 | 0.1629 |
| PSII reaction center PsbW | | POPTR_0002s25810g | 0.27 | 0.0149 | 0.36 | 0.0000 | 0.76 | 0.3453 |
| PSII assembly factor HCF136 | HCF136 | POPTR_0007s07780g | 0.22 | 0.0000 | 0.30 | 0.0000 | 0.75 | 0.2693 |
| Divinyl protochlorophyllide a 8-vinyl reductase | PCB2 | POPTR_0008s20940g | 0.24 | 0.0001 | 0.33 | 0.0000 | 0.71 | 0.2143 |
| Chlorophyll a/b-binding protein | LHCB3 | POPTR_0002s22220g | 0.12 | 0.0006 | 0.24 | 0.0000 | 0.50 | 0.0125 |
| Oxygen evolving enhancer 3 (PsbQ) | | POPTR_0010s17400g | 0.29 | 0.0005 | 0.38 | 0.0000 | 0.76 | 0.2936 |
| Cytochrome c biogenesis protein | | POPTR_0009s11490g | 0.36 | 0.0074 | 0.32 | 0.0000 | 1.14 | 0.6612 |
| Ferredoxin | | POPTR_0007s09630g | 0.10 | 0.0000 | 0.21 | 0.0001 | 0.46 | 0.0499 |
| Ferredoxin-thioredoxin reductase | | POPTR_0001s34690g | 0.32 | 0.0014 | 0.37 | 0.0000 | 0.86 | 0.5290 |
| Fructose-1,6-bisphosphatase | | POPTR_0005s21360g | 0.36 | 0.0032 | 0.52 | 0.0041 | 0.68 | 0.1035 |
| Starch synthase | | POPTR_0011s15540g | 0.03 | 0.0001 | 0.15 | 0.0001 | 0.22 | 0.1285 |

Ratio ≥2 or ≤0.5 and *P* value < 0.05 indicate a significant difference.

## 4. Discussion

In the present study, we found that the significantly up-regulated DEGs in oviposited leaves of '108' and '111' plants involved genes related to PR proteins such as TPIN and PR4/HEL. The soluble PR protein family consists of 17 PR family classifications, most of which have been shown to be rapidly induced both locally and systematically [34]. The specific defense functions of protease inhibitors (PR-6 family), phytodefense (PR-12 family), and lipid transfer protein (PR-14 family) in *A. thaliana* have been described in detail [35]. Plant PINs—anti-digestive proteins that affect the further feeding of insects by inhibiting the activity of serine proteases in the insect digestive tract [36]—can improve defenses against insects and pathogens [23]. Usually, damage-induced PIN genes include those encoding cysteine protease inhibitor (CPIN), serine protease inhibitor (SPIN), and insulin protease inhibitor (IPIN) [37,38]. Insect feeding can induce the expression of PIN genes in maize, potato, rice, and tomato plants [39]; moreover, expression of a class of mRNA similar to TPIN is induced when poplar is damaged [40]. In poplar trees, the protective activity of proteins including polyphenol oxidase (PPOs), endochitinase, and PINs induced against various pests has been well studied [41]. Meanwhile, although research on PR4/HEL is relatively lacking, it is considered to be a plant lectin with antibacterial activity [42]. PR4/HEL in the leaves of *Broussonetia papyrifera*, syn. *Morus papyrifera* (L.) have also exhibited antifungal activity [43]. Taking these findings into consideration, the oviposited '111' plants appeared to initiate an antimicrobial defense, which is consistent with the notion that insect eggs may be considered microbial pathogens [26].

Another significantly up-regulated gene in the oviposited '108' plants was NIMIN-2; while NIMIN-2 expression in the neighboring plants was up-regulated, there was no significant difference compared with that of the control. NIMIN2 proteins mediate pathogenesis-related protein 1 (PR-1) expression in tobacco, which is achieved through transient PR-1 inhibition before systemic acquired resistance (SAR) is fully developed [44]. NPR1/NIM1 is a key regulator of SAR in *A. thaliana* [45]; NPR1 controls the induction of PR genes for synthesizing SA and plays a crucial role in linking the JA and SA pathways [46]. The RPW8 homolog HR4 was also significantly up-regulated in the oviposited '108' plants. Up-regulated expression of RPW8—an atypical resistance protein and an important gene in activating basic resistance to powdery mildew [47]—in *A. thaliana* may enhance the basic defense ability of plants in an SA-dependent manner [48]. Metalloproteinases and L-ascorbate oxidase were also significantly up-regulated in oviposited '108' plants. PPO, a copper ion-dependent metalloproteinase, is an important defense protein that is critical in the defense against pathogenic bacteria and phytophagous insects [49]. Indeed, metalloproteinase activity was found to increase in the infection zone of silverleaf-susceptible plants and to affect the growth of *Chondrostereum purpureum* (Pers.), a fungus plant pathogen [50]. Ascorbate oxidase has been proposed to function as a plant defense protein that decreases the availability of ascorbate to insects [51]. Under stress, *Capsicum annuum* (L.) was found to increase the expression and activity of ascorbate oxidase [52]. Genes related to biological stress response in the neighboring plants, such as pathogen-responsive α-dioxygenase, were significantly up-regulated compared with those in the control and oviposited plants. α-Dioxygenase catalyzes the oxidation of fatty acids to produce a new group of oxidative lipids that protect tissues from oxidative damage and cell death and regulate the response to environmental stress in tomato roots [53].

In the '111' clone, PAD4 was significantly up-regulated in the oviposited plants. PAD4 is required for resistance against the eukaryotic biotroph *Hyaloperonospora parasitica* (Gaum.) [54]. PAD4 has a unique function in the innate immune response of plants, where it kills local cells in contact with lesions in *A. thaliana* [55]. BAP1 was another significantly up-regulated gene in oviposited '111' plants; however, it was significantly down-regulated in neighboring plants. BAP1 may play a direct role in regulating cell proliferation in wild *A. thaliana* [56] and it has been found that overexpression of BAP1 can lead to increased sensitivity to poisonous fungi, thereby negatively regulating plant defense responses [57]. These findings indicate that the oviposited and neighboring '111' plants activate different defense mechanisms to regulate the *BAP1* gene. The expression of XTR8, an enzyme that causes plant cell wall-loosening for cell expansion [58], was significantly up-regulated in oviposited

and neighboring '111' plants. The activity of this enzyme is closely related to the formation of secondary cell walls in the poplar stem [59]. GO functional enrichment analysis indicated that the DEGs of oviposited and neighboring '111' plants were mainly involved in the "cell part" function, which is consistent with XTR8 participating in defense function in "cell part".

Antibacterial genes were up-regulated in both the oviposited and neighboring '108' and '111' plants, which may have activated the SA-mediated defense system. Studies have found that the mass accumulation of SA under deposited eggs can induce the expression of many SA-responsive genes [21,27]. The importance of the oviposition-induced SA pathway in *A. thaliana* has also been demonstrated [28]. However, the interaction between trees and insects is more complex and variable, as trees may have more signal transduction pathways than herbaceous plants [60]. In addition, recent studies have found that poplar-specific genes are significantly enriched in translation and energy metabolism [61]. More evidence is needed to validate the signaling pathways that induce resistance in woody plants by oviposition. The oxidation reaction in oviposited leaves and the signal conduction between the "oviposition exciter" and oviposited leaves will be studied in the future. Meanwhile, cytochrome P450, associated with the tryptophan pathway, was significantly up-regulated in the neighboring '111' plants. Cytochrome P450 plays an important role in protecting organisms from external invasion and is widely involved in the production of secondary metabolites [62]. Indeed, enrichment of metabolic pathways and secondary metabolites in the neighboring plants of '111' was the highest. The formation of secondary metabolites is generally considered a manifestation of plant resistance. For example, levels of tannins—compounds that affect insect digestion of protein and starch [63]—are generally higher in insect-resistant varieties of poplar than in nonresistant varieties. Phenolic compounds can be condensed into tannin and lignin, and their composition and content are closely associated with plant resistance against insects [64,65]. For instance, the higher the phenol and phenolic acid content in poplars, the stronger the resistance to *Anoplophora glabripennis* (Mot.) (Coleoptera: Cerambycidae), an Asian long-horned beetle [66]. Studies have found that the resistance of different clones of *Populus deltoids* to *Clostera anastomosis* L. (Lepidoptera: Notodontidae) is positively correlated with the total phenol content in leaves [67]. The above findings indicate that the neighboring '111' plants increased their resistance against *M. sieversi* through the synthesis of secondary metabolites.

*Pinus sylvestris* (L.) reduces photosynthetic activity after oviposition by *Diprion pini* (L.), while genes associated with photosynthesis are suppressed [7]. Photosynthesis of cabbage leaves is also inhibited after oviposition by *Murgantia histrionica* (Hahn) (Heteroptera: Pentatomidae) [68]. This reduction in photosynthesis occurs because the defense responses consume much energy [69]. Similarly, in the present study, genes associated with photosynthetic activity and carbohydrate biosynthesis processes were down-regulated in the oviposited and neighboring '108' and '111' plants. Thus, these plants increase their resistance against insects by reducing photosynthesis and redirecting their energy expenditure.

Overall, our findings indicate that oviposited and neighboring plants show differential expression of the resistance genes in response to oviposition by *M. sieversi*. Further RT-PCR will also be performed in the follow-up study, to validate the defense gene discovered (and reported) in the present study. We believe that the differences of resistance between the two *Populus* section *Aigeiros* clones against *M. sieversi*, as well as the mechanism of oviposition-induced resistance will be further revealed in our other work.

## 5. Conclusions

The present study demonstrated via transcriptome analysis that with different resistance mechanisms, both oviposited and neighboring '108' and '111' clones of *Populus* section *Aigeiros* induce resistance against *M. sieversi* to varying degrees. The DEGs in oviposited and neighboring plants were mainly involved in the "cell part", "catalysis", and "metabolic process" functions, but the metabolic pathways that were enriched were different. Although antibacterial genes were up-regulated in both the oviposited and neighboring '108' and '111' plants, there were

differences in specific genes. Moreover, synthesis of metabolic pathways and secondary metabolites showed the highest enrichment, whereas nutrient accumulation, such as for photosynthesis and carbohydrate synthesis, were significantly down-regulated. Overall, our findings indicate that oviposited and neighboring plants show differential expression of resistance genes in response to oviposition by *M. sieversi*. Our study provides insights into the molecular resistance responses of woody plants upon oviposition by herbivorous insects. Further research on the signal conduction between the "oviposition exciter" and oviposited leaves may help cultivate insect-resistant poplars in the future.

**Author Contributions:** Conceptualization, L.G. and Z.Z.; methodology, L.G. and Z.Z.; investigation, L.G. and F.L.; formal analysis, S.Z. and X.K.; writing—original draft, L.G.; writing—review and editing, L.G.; S.Z. All authors have read and agreed to the published version of the manuscript.

**Funding:** This research was supported by the Special Fund for Forest Scientific Research in the Public Welfare, grant number 201504302.

**Acknowledgments:** We thank the staff of Xuzhou Forest Station in Shandong Province, China for collecting pupae of *M. sieversi* (Staudinger).

**Conflicts of Interest:** The authors declare no conflict of interest.

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
