# Peer review of "Effect of Micromelalopha sieversi (Staudinger) Oviposition Behavior on the Transcriptome of Two Populus Section Aigeiros Clones"

_forests, doi:10.3390/f11091021_

Round 1

Reviewer 1 Report

The authors responded to my comments. Overall, they corrected the manuscript correctly in most cases, however I have the impression that they made some "cosmetic" changes. Also, a few comments were omitted. I still recommend shortening the introduction chapter (look at the first two paragraphs) and simplifying the description of the results.

Clearly defined goal and hypotheses at the end of the introduction are also necesarry.

At the end of the discussion, I would suggest adding a few sentences to the potential limitations of this research.

Author Response

Point 1: The authors responded to my comments. Overall, they corrected the manuscript correctly in most cases, however I have the impression that they made some "cosmetic" changes. Also, a few comments were omitted. I still recommend shortening the introduction chapter (look at the first two paragraphs) and simplifying the description of the results.

Response 1: Thank you very much for your advice. The first paragraph of the introduction has been abridged and part of the literature was removed, including L41-48 and L55-56. The second paragraph of the introduction was a review of the literature on the defense gene expression of the plant oviposited by insect, which was simply modified and the sequence of the literature has been adjusted. In this study, the plants were divided into the following treatment groups: oviposited, neighboring, and control leaves of two Populus section Aigeiros clones (“108” and “111”) (108-O, 108-N, 108-C and 111-O, 111-N, 111-C, respectively). Each clone was compared between two treatments, so the results were more content, has been simplified as far as possible.

Clearly defined goal and hypotheses at the end of the introduction are also necesarry.

Response 2: Thank you very much for your advice. At the end of the introduction, the research purpose of this experiment has been supplemented, L97-101.

At the end of the discussion, I would suggest adding a few sentences to the potential limitations of this research.

Response 3: Thank you very much for your advice. At the end of the discussion, the follow-up study has been described, L418-423.

Reviewer 2 Report

Dear all, I am satisfied about the answers authors gave to me so I should accept the revision.

Author Response

Point 1: Dear all, I am satisfied about the answers authors gave to me so I should accept the revision.

Response 1: Thank you for the remark.

This manuscript is a resubmission of an earlier submission. The following is a list of the peer review reports and author responses from that submission.

Round 1

Reviewer 1 Report

Dear authors and editor. I am pleased to have reviewed your MS. The topic of your work is very interesting but extremely ambitious. Transcriptome analysis technically seems ok but I doubt about the results. I observed a variation in gene expression among two varieties even in the controls. Trees and plants do not have a strict gene expression pattern during their development and changes can be observed even among same leaves of the same plant or different parts of sampling among the same leaf. As for control I would expect to use leaves that were artificially oviposited with maybe a droplet of water or another chemical. Also I would expect a bioassay to confirm differential gene expression by using RT-PCR. Τranscriptome is not enough. 

Reviewer 2 Report

The reviewed work is very interesting and contains a lot of results that could be successfully included in two separate manuscripts. I understand the authors' approach, because now all the content is consistent, although I miss a clearly defined goal and hypotheses at the end of the introduction. However, I still recommend shortening the introduction chapter (look at the first two paragraphs) and simplifying the description of the results. Authors should also verify that all citations are necessary.

Furthermore, I have a methodological objections: why was the control variant performed in a different laboratory (L118) ???
Another serious problemis the statistical approach. Why, for example in Tables 3 and 4, do you use pairwise analysis? I think a two-way ANOVA would be a better solution (influence of species, treatment and their interactions).

Below, I present the smaller shortcomings:
1. You are not consistent in naming of species. The correct name should also include the name of the discoverer (as it did in L 361). L110: don't start a sentence with an abbreviation.
2. Table 1 description: I miss here an explanation of the variant abbreviations. They appear somewhere later, however the reader should already have them presented and explained here.
3. Table 2: did you miss "enter" at the end of the table ??
4. Fig 1. There is no good description of the legend; the font could be larger.
5. L198 delete the line on the right
6. L245 add enter
7. L246-L249 try to enlarge the font.

Overall, I think the work is good and deserves to be published. It should first be corrected by the authors.